# How to GAN Event Unweighting

**Mathias Backes, Anja Butter,**
**Tilman Plehn and Ramon Winterhalder***

Institut für Theoretische Physik, Universität Heidelberg, Germany

* winterhalder@thphys.uni-heidelberg.de

## Abstract

Event generation with neural networks has seen significant progress recently. The big open question is still how such new methods will accelerate LHC simulations to the level required by upcoming LHC runs. We target a known bottleneck of standard simulations and show how their unweighting procedure can be improved by generative networks. This can, potentially, lead to a very significant gain in simulation speed.

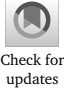

**Content**

## 1 Introduction

First-principle simulations have defined data analysis at the LHC since its beginning. The success of the LHC in establishing the Standard Model as the fundamental theory of particle interactions is largely owed to such precision simulations and the qualitative progress in our understanding of QCD. Because the HL-LHC will produce a data set more than 25 times the current Run 2 data set, the current theory challenge is to provide significantly faster simulations, while at the same time increasing the precision to the per-cent level and better. This

goal is defined by QCD precision predictions as well as by the expected size of experimental uncertainties, which seriously limit the use of leading-order simulations even for complex signatures at future LHC runs. While it is hard to accelerate standard tools to the required level, there is justified hope that modern machine learning will allow us to reach this goal.

A range of modern neural network applications to LHC simulations have been proposed over the last two years [1]. The conceptually most ambitious network architecture are generative networks, like generative adversarial networks (GAN) [2–4], variational autoencoders [5, 6], or normalizing flows [7–9] including invertible neural networks (INNs) [10–12]. Following the established Monte Carlo structures leads us to consider phase space integration [13, 14], phase space sampling [15–18], and amplitude networks [19,20]. A technical step beyond the standard approach are fully network-based event generation [21–26], including event subtraction [27], detector simulation [28–37], or parton showering [38–42]. Generative networks can also help to extract new physics signals [43] or experimental anomalies [44,45]. Conceptually more far-reaching, conditional generative networks can invert the forward simulation chain to unfold detector effects [46,47] and extract the hard scattering at parton level [48]. Many of these applications are currently finding their way into the LHC theory toolbox.

Going back to the LHC motivation, the key question is where we can gain significant speed in precision theory simulations. As mentioned above, we can use flow networks to improve the phase space sampling [15, 17]. In addition, we can employ generative networks because they learn more information than a statistically limited training data set [4]. This is why neural networks are successfully used to encode parton densities [49]. Finally, there exist promising hints for network extrapolation in jet kinematics [23].

In this paper we follow a different path and target a well-known bottleneck LHC event simulation, the transformation of weighted into unweighted events [50, 51]. Usually, the information about a differential scattering rate is first encoded in a combination of event weights and the event distribution over phase space. To compare with data we ideally work with unit-weight events, where all information is encoded in the event distribution. For complex processes, the standard unweighting procedures suffer from low efficiency, which means they lose statistical power. We will show how a generative network, specifically a GAN, can unweight events without these losses and thereby speed up event generation significantly [52]. We will start with a 1-dimensional and a 2-dimensional toy model in Sec. 2, to illustrate our uwGAN idea in the context of standard approaches. In Sec. 3 we will then use a simple LHC application to show how our GAN-unweighting method can be applied to LHC simulations.

## 2 Unweighting GAN

Before we show how networks can be useful for LHC simulations, we briefly introduce event unweighting as it is usually done, how a generative network can be used for this purpose, and when such a network can beat standard approaches. First, we will use a 1-dimensional camel distribution to illustrate the loss function which is needed to capture the event weights. Second, we use a 2-dimensional Gaussian ring as a simple example where our method circumvents known challenges of standard tools.

---

in short GANweighting or GUNweighting.

## 2.1 Unweighting

For illustration purpose, we consider an integrated cross section of the form

$$\sigma = \int dx \, \frac{d\sigma}{dx} \equiv \int dx \, w(x) \,, \tag{1}$$

where $d\sigma/dx$ is the differential cross section over the $m$-dimensional phase space $x$. To compute this integral numerically we draw $N$ phase space points or events $\{x\}$ and evaluate

$$\sigma \approx \left\langle \frac{d\sigma}{dx} \right\rangle \equiv \langle w(x) \rangle \,. \tag{2}$$

The event weight $w(x)$ describes the probability for a single event $x$. Sampling $N$ phase space points $\{x\}$ and evaluating their weights $\{w\}$ defines $N$ weighted events $\{x, w\}$. The information on the scattering process is encoded in a combination of event weights and phase space density. This can be useful for theory computations, but actual events come with unit weights, so all information is encoded in their phase space density alone.

We can easily transform $N$ weighted events $\{x, w\}$ into $M$ unweighted events $\{x\}$ using a hit-or-miss algorithm, where in practice $M \ll N$. It re-scales the weight $w$ into a probability to keep or reject the event $x$,

$$w_{\text{rel}} = \frac{w}{w_{\text{max}}} \,, \tag{3}$$

and then uses a random number $R \in [0, 1]$ such that the event is kept if $w_{\text{rel}} > R$. The obvious shortcoming of this method is that we lose a lot of events. For a given event sample the unweighting efficiency is [15]

$$\epsilon_{\text{uw}} = \frac{\langle w \rangle}{w_{\text{max}}} \,. \tag{4}$$

If the differential cross section varies strongly, $\langle w \rangle \ll w_{\text{max}}$, this efficiency is small and the LHC simulation becomes CPU-intensive.

A standard method to improve the sampling and integration are phase space mappings, or coordinate transformations $x \to y(x)$,

$$\sigma = \int dx \, w(x) = \int dy \left| \frac{\partial x}{\partial y} \right| w(y) \equiv \int dy \, \tilde{w}(y) \,. \tag{5}$$

Ideally, the new integrand $\tilde{w}(y)$ is nearly constant and the structures in $w(x)$ are fully absorbed by the Jacobian. In that case

$$\tilde{\epsilon}_{\text{uw}} = \frac{\langle \tilde{w} \rangle}{\tilde{w}_{\text{max}}} \approx \frac{\langle C \rangle}{C} = 1 \,. \tag{6}$$

This method of choosing an adequate coordinate transformation is called importance sampling. The most frequently used algorithm is Vegas [53,54], which assumes that $g(x)$ factorizes into phase space directions, as we will discuss later.

In contrast to vetoing most of the weighted events we propose to use all of them to train a generative model to produce unweighted events. We follow a standard GAN setup with spectral normalization [55] as regularization method

$$\begin{aligned} L_D &= \left\langle -\log D(x) \right\rangle_{x \sim P_T} + \left\langle -\log(1 - D(x)) \right\rangle_{x \sim P_G} \,, \\ L_G &= \left\langle -\log D(x) \right\rangle_{x \sim P_G} \,. \end{aligned} \tag{7}$$

For weighted training events, the information in the true distribution $P_T$ factorizes into the distribution of sampled events $Q_T$ and their weights $w(x)$. To capture this combined information we replace the expectation values by weighted means for batches of weighted events,

$$
\begin{aligned}
L_D^{(\text{uw})} &= \frac{\langle -w(x) \log D(x) \rangle_{x \sim Q_T}}{\langle w(x) \rangle_{x \sim Q_T}} + \langle -\log(1 - D(x)) \rangle_{x \sim P_G}, \\
L_G^{(\text{uw})} &= \langle -\log D(x) \rangle_{x \sim P_G}.
\end{aligned}
\tag{8}
$$

Because the generator produces unweighted events with $w_G(x) = 1$ their weighted mean reduces to the standard expectation value. This way, our unweighting GAN (uwGAN) unweights events, and the standard GAN is just a special case with all information encoded in the event distribution.

## 2.2 One-dimensional camel back

We illustrate the unweighting GAN with a 1-dimensional camel back

$$
P_{\text{camel}}(x) = 0.3 \, \mathcal{N}(x; \mu = 0, \sigma = 0.5) + 0.7 \, \mathcal{N}(x; \mu = 2, \sigma = 0.5),
\tag{9}
$$

where $\mathcal{N}(x; \mu, \sigma)$ is a Gaussian. To see how the GAN reacts to different ways of spreading the information between weights and distributions, we define three events samples,

· unweighted events distributed according to the camel back, $X_u = (x_{\text{camel}}, w_{\text{uniform}})$;

· uniformly distributed events $X_w = (x_{\text{uniform}}, w_{\text{camel}})$; and

· a split $X_{\text{hybrid}} = (x_{q_1}, w_{q_2})$ with $P_{\text{camel}}(x) \propto q_1(x) q_2(x)$ and $q_1(x) = \mathcal{N}(x; \mu = 0, \sigma = 1)$.

For the camel back example our training data will consist of 1 million weighted events. We use 32 units within 2 layers in the generator and 32 units within 3 layers in the discriminator. In the hidden layers, we employ the standard ReLU activation function, $\max(0, x)$, for both networks. To compensate an imbalance in the training we update the discriminator ten times as often as the generator. As a first test, we show in Fig. 1 how our GAN reproduces the full 1-dimensional target distribution from unweighted events, uniformly distributed events, and weighted events equally well. The limitation are always the poorly populated tails in the training data.

To benchmark our unweighting GAN, we first sample the true distribution with a large number of events and bin them finely, in our 1-dimensional case $10^{10}$ events in 2000 bins equally distributed over the full range $x = -2 \ldots 4$. This statistics goes far beyond the training sample and is only needed to define a truth benchmark. We then generate an equally large sample of $10^{10}$ GAN events and compare the two high-statistics samples, weighted truth events and unweighted GANned events, in the top panels of Fig. 2. From the bin-wise ratio we see that the GAN reproduces the true distribution at the few per-cent level, again limited by the tails.

Given the true distribution, we can compute event-wise factors which would be needed to shift the GANned unit weights to reproduce the true distribution exactly. We refer to them as truth-correction weights for each (unweighted) GAN event. Because we rely on the binned truth information we assign the same truth correction to all GAN events in a given, narrow bin. Formally, we assume that the generator distribution $P_G$ approximates the true distribution $P_T$, so the bin-wise ratio

$$
w_G(x) = \frac{P_T(x)}{P_G(x)}
\tag{10}
$$

for each unweighted event, given its phase space position $x$, should tend to one. The actual values for the truth-correction weights are shown in the bottom panels of Fig. 2. For the full

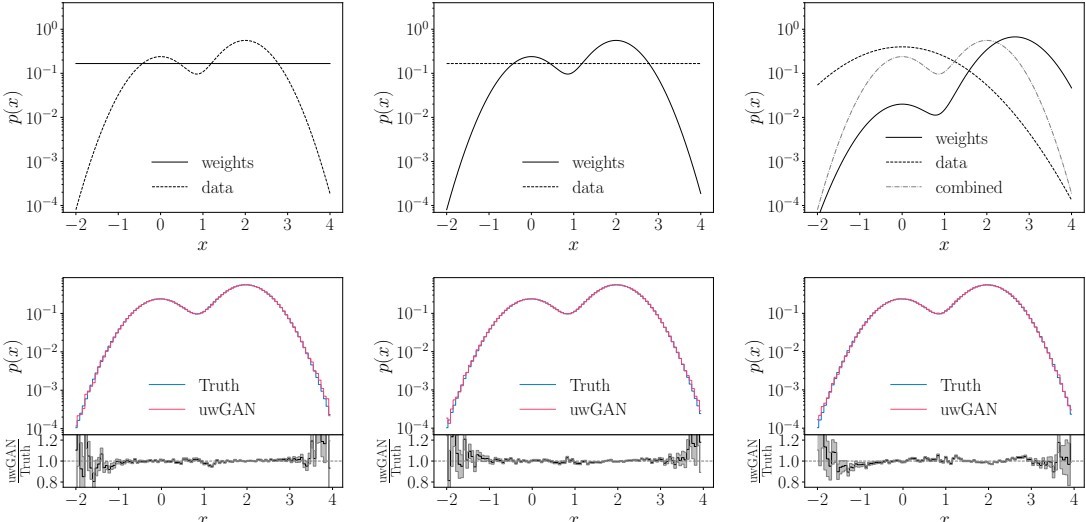

Figure 1: Event vs weight distribution of the training data (top) and GANned vs truth kinematic distributions (bottom) for unweighted events (left), uniform distribution (center), and our hybrid case (right). The lower panels in the bottom show the bin-wise truth-to-GAN ratio.

$x$-range we see that they are strongly peaked around unity, but with sizeable tails. The fact that the distribution is not symmetric and includes significant statistical fluctuations suggests that our network could be further improved. Nevertheless, the vast majority of events have a truth-correction below 3%. In the right panel we see the same distribution after removing the tails. Literally all GAN events now come with a truth-correction below 3%. Comparing the upper and lower panels of Fig. 2 we also see that these truth-correction weights are not statistically distributed corrections, fluctuating rapidly as a function of $x$. Instead, they reflect systematic limitations to the precision with which the GAN learns $P_T(x)$ and encodes it into the phase space distribution.

As discussed above, `Vegas` encodes $P_T(x)$ jointly into the phase space distribution and event weights [53, 54]. This means we can compare the GAN and `Vegas` encodings in the phase space distribution by comparing the truth-correction weights in the sense that for `Vegas` they will define the perfectly trained output. After a series of 150 adaption steps, `Vegas` reaches the weight distribution shown in Fig. 2, corresponding to an unweighting efficiency of 0.75. Note that after 50 adaption steps, this `Vegas` unweighting efficiency was 0.95. The reason is that `Vegas` is optimized for integration by using tight grids in the bulk and wide grids in the tails. The longer `Vegas` adapts its grid, the more events are removed from the tails. This improves the numerical integration at the cost of the unweighting efficiency. Indeed, in Fig. 2 we see that the high-weight tails of the `Vegas` truth-correction are comparable to the GAN case. Again the tails in the event weights correspond directly to the tails of the density distribution over $x$. When it comes to unweighting the `Vegas` events, these tails become a major problem, because they drive the denominator in Eq.(4).

## 2.3 Two-dimensional Gaussian ring

Knowing a weakness of `Vegas` we now choose a 2-dimensional circle in the $x$-$y$ plane with a Gaussian radial distribution as our second example,

$$P_{\text{circle}}(x,y) = N \exp\left[-\frac{1}{2\sigma^2}\left(\sqrt{(x-x_0)^2 + (y-y_0)^2} - r_0\right)^2\right], \tag{11}$$

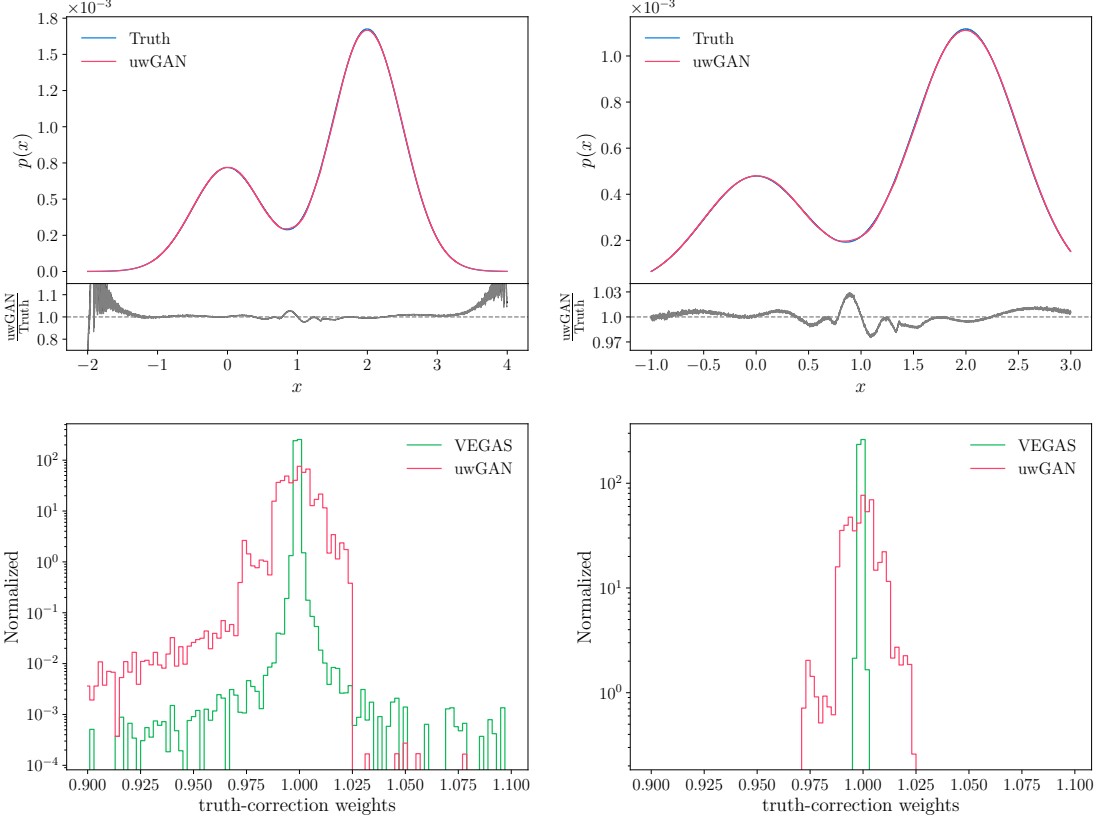

Figure 2: Upper: training and GANned distribution for the 1-dimensional camel back. In the right panels we remove the tails of the distribution. Lower: truth-correction weights for the GANned events, compared with the `Vegas` weight distribution.

with $x_0 = y_0 = 0.5$, $r_0 = 0.25$, and $\sigma = 0.05$. The normalization is then given by $N \approx 5.079$. We use the same GAN architecture as before, but with 256 units in 8 layers in, both, generator and discriminator.

In Fig. 3 we show the true distribution as well as the asymmetry of the truth and GANned distributions. As for the 1-dimensional camel back, large relative deviations are limited to the tail of the distribution, in this case including the center of the circle. In the lower-left panel of Fig. 3 we see how these regions contribute little to the integral over the density.

It is clear that the `Vegas` algorithm cannot reproduce the circular shape, because it breaks the factorization with the dimensionality. Instead, `Vegas` constructs a square with a low unweighting efficiency. Again, we compare the GAN and `Vegas` truth-correction weights in the lower-right panel of Fig. 3. As expected, the uwGAN now does significantly better, albeit with truth corrections up to ±25% in the tails. Just like for the 1-dimensional example, the tails in the truth-correction correspond directly to the tails in the density, so they reflect the statistical limitations of the training sample. For a realistic application the key question becomes how this kind of truth correction compares to the standard approaches and if it is sufficient given the general statistical limitations in poorly populated phase space regions.

As a side remark, it is of course possible to compute the truth corrections without binning for the 1-dimensional and 2-dimensional toy models. However, for a realistic LHC problem that will in general not be the case, so we stick to the binned definition throughout this paper. We have explicitly tested that our binned distributions agree with the exact truth-correction distributions for the two toy models.

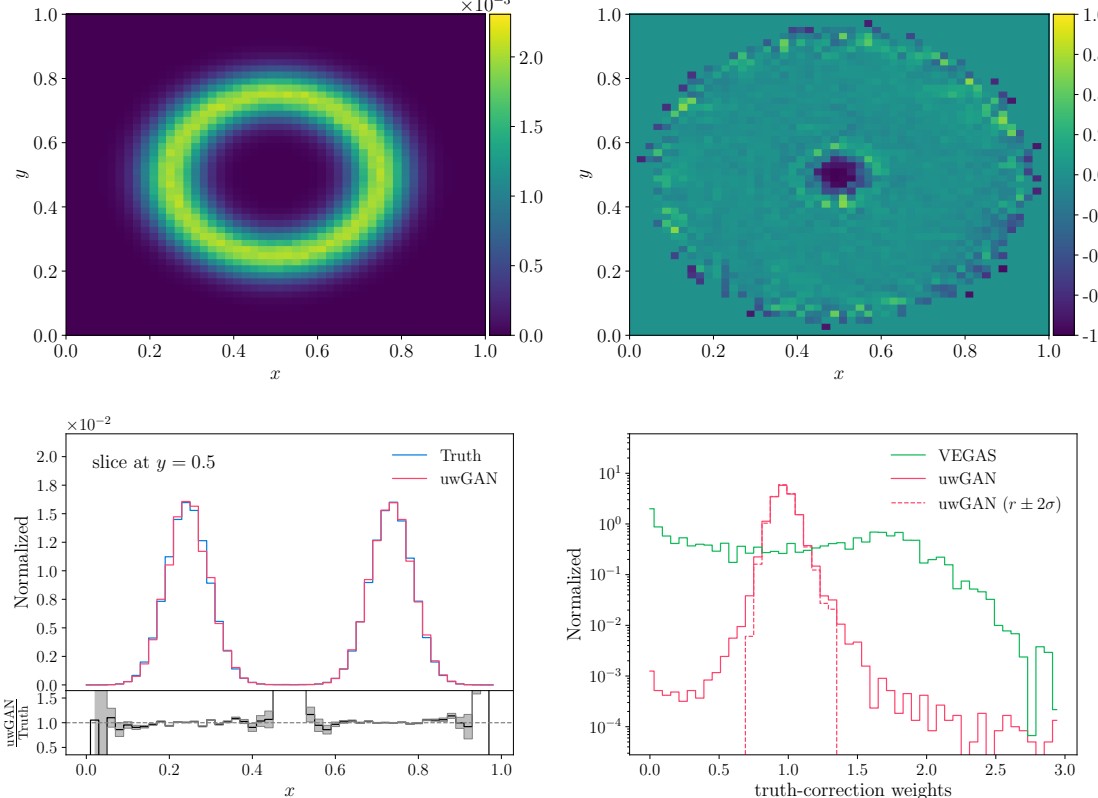

Figure 3: Results for the 2-dimensional Gaussian ring showing the truth data (upper left), the asymmetry between truth and uwGAN (upper right), a 1-dimensional slice at $y = 0.5$ (lower left), and the comparison of the truth-correction for uwGANned events with Vegas events (lower right). The dotted curve includes the bulk region $r \in [r_0 - 2\sigma, r_0 + 2\sigma]$ only.

## 3 Unweighting Drell–Yan

So far, we have considered two toy examples to motivate our uwGAN. Next, we need to apply it to a simple LHC process, where we can study the phase space patterns in some detail. We consider the Drell–Yan process

$$pp \rightarrow \mu^+ \mu^- . \tag{12}$$

We generate 500k weighted events at a CM energy of 14 TeV. The 4-dimensional fiducial phase space is defined by the minimal acceptance cut

$$m_{\mu\mu} > 50 \text{ GeV} \tag{13}$$

to avoid the photon pole in the numerical event generation. The technical requirement on the weighted training events is that they should cover a wide range of weights, so we can test if the uwGAN can deal with this practical challenge. This means we cannot use a standard Monte Carlo, where sophisticated phase space mappings encode $p_T$ and $m_{\mu\mu}$ very well.

We implement our own custom event generator in Python, extracting the matrix elements from Sherpa [56], the parton densities from LHAPDF [57], and employing the Rambo-on-diet sampling [58, 59]. The integration over the parton momentum fractions is symmetrized in

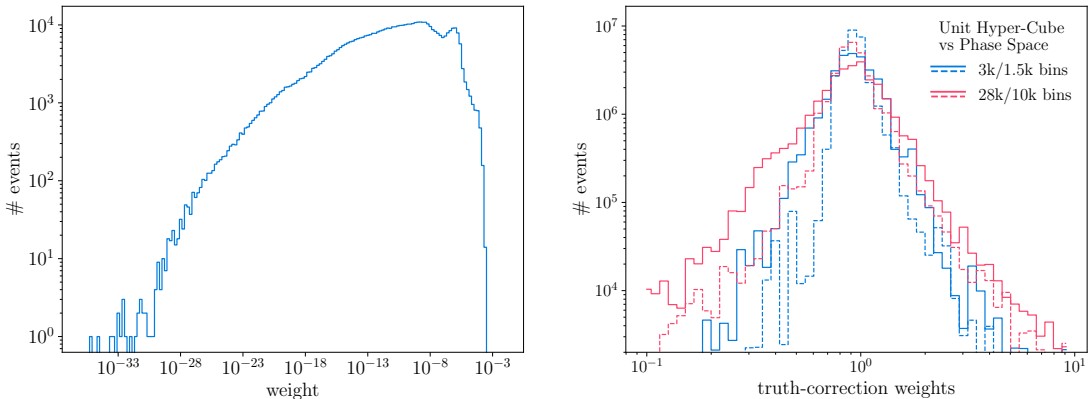

Figure 4: Left: Weight distribution for 500k weighted training events. Right: Truth-correction weights for 30M uwGAN events on the unit hyper-cube (solid) and the phase space parametrization of Eq.(17) (dashed) for 2k (blue) and 14k (red) bins in 4-dimensions.

terms of $\tau = x_1 x_2$ as the first phase-space variable,

$$\sigma = \int_{\tau_{\min}}^{1} \frac{d\tau}{\tau} \int_{\tau}^{1} \frac{dx_1}{x_1} \sum_{a,b} x_1 f_a(x_1)\, x_2 f_b(x_2)\, \hat{\sigma}_{ab}(x_1 x_2 s)\,, \tag{14}$$

with Eq.(13) translating into $\tau_{\min} \approx 0.00128$. Mapping the phase space onto a unit hyper-cube defines two random numbers $r_{1,2}$ through

$$\tau = \tau_{\min}^{r_1} \qquad x_1 = \tau^{r_2} \qquad x_2 = \tau^{1-r_2}\,, \tag{15}$$

such that

$$\sigma = 2\log\tau_{\min} \int_0^1 dr_1\, r_1 \int_0^1 dr_2 \sum_{a,b} x_1 f_a(x_1)\, x_2 f_b(x_2)\, \hat{\sigma}_{ab}(x_1 x_2 s)\,. \tag{16}$$

With an additional random number $r_3 = (\cos\theta + 1)/2$ we can parametrize the 4-dimensional phase space as

$$
\begin{aligned}
p_T &= 2 E_{\text{beam}}\, \tau_{\min}^{r_1/2} \sqrt{r_3(1-r_3)} \\
p_{z_1} &= E_{\text{beam}} \left( \tau_{\min}^{r_1 r_2} r_3 + \tau_{\min}^{r_1(1-r_2)}(r_3 - 1) \right) \\
p_{z_2} &= E_{\text{beam}} \left( \tau_{\min}^{r_1 r_2}(1 - r_3) - \tau_{\min}^{r_1(1-r_2)} r_3 \right) \\
\phi &= 2\pi r_4\,.
\end{aligned}
\tag{17}
$$

In Fig. 4 we show the weight distribution for our event generator, where the shown 500k event weights are computed as the product of scattering amplitude, parton density, and phase-space factor. While the distribution is very smooth, indicating that the phase space is sampled precisely, the range of weights poses a problem for an efficient event unweighting. Even if we are willing to ignore more than 0.1% of the generated events, we still need to deal with event weights from $10^{-30}$ to $10^{-4}$. Effects contributing to this vast range are the $Z$-peak, the strongly dropping $p_T$-distributions, and our deliberately poor phase space mapping. The classic unweighting efficiency defined by Eq.(4) is 0.22%, which is considered high for state-of-the-art

tools applied to complex LHC processes. In the following panels of Fig. 5 we show a set of kinematic distributions, first for the 500k weighted training events including the deviation from a high-precision truth sample. Indeed, this training data-set describes $E_\mu$ all the way to 6 TeV and $m_{\mu\mu}$ beyond 250 GeV with deviations below 5%. The perfectly flat $\phi_\mu$ distribution turns out to be the challenge in our specific phase space parametrization, with bin-wise deviations of up to 20% from the true distribution.

In addition to the unweighted training data, we also show the kinematic distributions for unweighted events from a standard algorithm. We use the hit-and-miss method described in Sec. 2.1 without any further improvements, which limits the number of unweighted events to 1000. Correspondingly, the standard unweighted events only cover $E_\mu$ to 1 TeV and $m_{\mu\mu}$ to 110 GeV. For $\phi_\mu$ the deviations also exceed those of the training data significantly. This poor behavior is simply an effect of the low unweighting efficiency and a serious challenge for LHC precision simulations.

Alternatively, we can employ our uwGAN to unweight the Drell-Yan training data. To take into account symmetries, we only generate the degrees of freedom of the process. By construction, this guarantees momentum conservation and on-shell conditions. Before passing to the discriminator both the generated batches $\{x_G\}$ and the truth batches $\{x_T\}$ are parameterized as

$$x = \{p_T, p_{z_1}, p_{z_2}, \phi, w\}, \tag{18}$$

where $w$ is the associated event weight. In order to reproduce the sharp resonance appearing in the $m_{\mu\mu}$ distribution which originates from the $Z$ boson propagator, we employ an additional maximum mean discrepancy (MMD) loss [24,60]. This loss ensures that the network learns a pre-defined low-dimensional function over the high-dimensional phase space. For the unweighting GAN we generalize it in analogy to Eq.(8),

$$L_{\text{wMMD}} = \left[ \frac{\langle w(x)\,w(x')\,k(x,x')\rangle_{x,x'\sim Q_T}}{\langle w(x)\,w(x')\rangle_{x,x'\sim Q_T}} + \langle k(y,y')\rangle_{y,y'\sim P_G} - 2\frac{\langle w(x)\,k(x,y)\rangle_{x\sim Q_T, y\sim P_G}}{\langle w(x)\rangle_{x\sim Q_T}} \right]^{\frac{1}{2}}, \tag{19}$$

Table 1: Details for our uwGAN setup for the Drell-Yan process.

| Parameter | Value |
|---|---:|
| Layers | 6 |
| Kernel initializer | He uniform |
| G units per layer | 414 |
| D units per layer | 187 |
| G activation function | ReLU |
| D activation function | leaky ReLU |
| D updates per G | 2 |
| $\lambda_{\text{wMMD}}$ | 2.37 |
| Learning rate | 0.0074 |
| Decay | 0.42 |
| Batch size | 1265 |
| Epochs | 500 |
| Iterations per epoch | 200 |

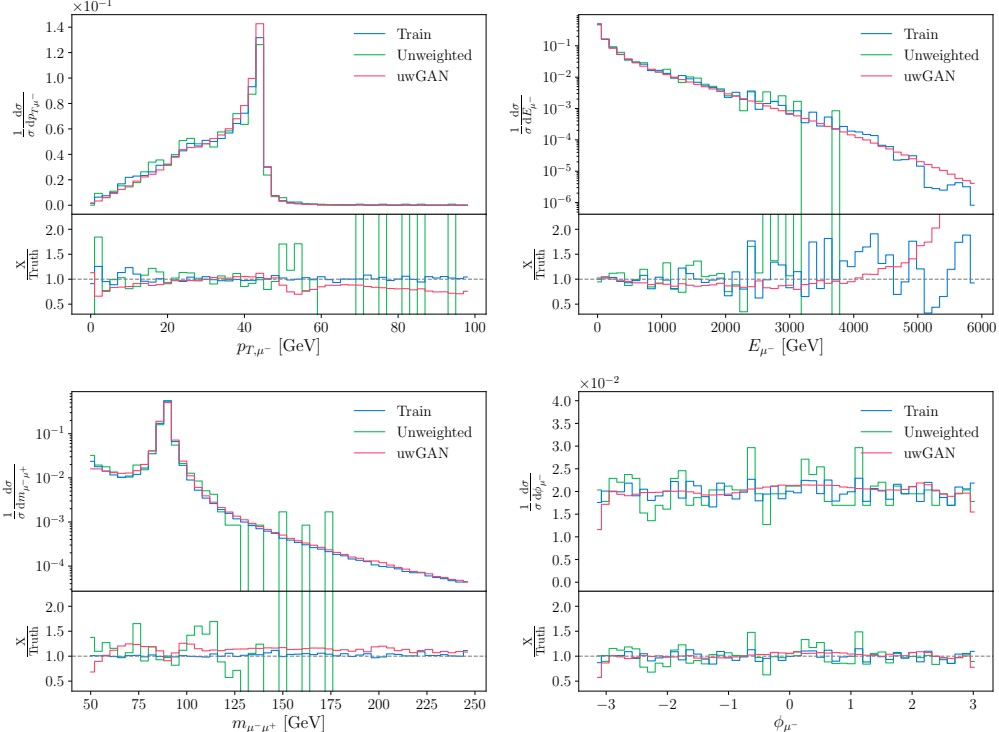

Figure 5: Exemplary kinematic distributions for the Drell-Yan process. For the kinematic distributions we show the 500k weighted training events, 1k unweighted events using the standard unweighting algorithm discussed in Sec. 2.1, and 30M uwGAN events.

where we already use that $w_G(y) = 1$. Note that we use MMD instead of MMD$^2$ as this increases the sensitivity of the loss close to zero. This loss is then added to the generator objective

$$L_G \rightarrow L_G + \lambda_{\mathrm{wMMD}} L_{\mathrm{wMMD}} . \tag{20}$$

The network parameters are given in Tab. 1. The parameters in the upper panel have been determined by a random hyperparameter search and have shown the best results.

In the right panel of Fig. 4 we again show the truth-correction weights for our uwGAN events, evaluated on the binned phase space either in terms of the unit hyper-cube ($r_j = 1 \dots 0$) or the appropriately cut phase space of Eq.(17). The number of bins ignores empty bins and shows the limitations of our bin-wise extraction of the truth correction. While some of the truth corrections are not negligible, we also know that they appear in the tails of the generated phase space distribution and can easily be traced. Even if we consider the finite and bin-wise-defined truth correction with a grain of numerical salt, we find the performance of our relatively slim network quite convincing, given that we start from weighted events with more than 25 orders of magnitude in weights. Most importantly, the tails of the truth correction are a result of the uwGAN unweighting, not a limiting factor like for the standard unweighting procedure.

The appropriate measure of success for our uwGAN are the predicted kinematic distributions. In Fig. 5 we compare the weighted training data, a corresponding unweighted event sample using the standard algorithm, and the uwGAN results. In the lower panels we show the relative differences to the truth, defined as a high-statistics version of the training sample. While the training data agrees with the truth very well, we see its statistical limitations in the tail of the $E_\mu$-distribution. In addition, the $\phi_\mu$ distribution for the weighted training data is noisier than one would expect for a smooth phase space.

In accordance with the established performance of GANs, the uwGANned events reproduce the truth information well. As always, the GAN learns the phase space information only to the point where it lacks training statistics and the GAN undershoots the true distribution [24]. This limitation can be quantitatively improved by using different network architectures [48]. In our case it affects the phase space coverage for $E_\mu \gtrsim 4.5$ TeV and $m_{\mu\mu} \gtrsim 250$ GeV. These values are not quite on par with the training data, but much better than for standard unweighting. For the same two distributions we clearly see the loss of information from standard hit-and-miss unweighting, and how the uwGAN avoid these large losses. Along the same line, the $\phi_\mu$ distribution shows how the uwGAN even slightly smoothes out the noisy training data. We can translate the reduced loss of information into a corresponding size of a hypothetical hit-and-miss training sample, for instance in terms of rate and required event numbers, and find up to a factor 100 for our simple example.

## 4 Outlook

First-principle precision simulations are a defining aspect of LHC physics and one of the main challenges in preparing for the upcoming LHC runs. Given the expected experimental uncertainties, we need to improve both, the precision and the speed of the theory-driven event generation, significantly to avoid theory becoming the limiting factor for the majority of LHC analyses. One promising avenue is modern machine learning concepts applied to LHC event generation.

In this study we proposed a significant improvement to one of the numerical bottlenecks in LHC event generation, the unweighting procedure. Such an unweighting step is part of every event generator, and for complex final state it rapidly becomes a limiting factor. We showed how to train a generative network on weighted events, with a loss function designed to generate events of unit weights, or unweighted events.

For a 1-dimensional and a 2-dimensional toy model we have shown that our uwGAN can indeed be used for event unweighting and that in the limit of perfect training it reproduces the true phase space distributions just like standard methods like `Vegas`. While we cannot beat the `Vegas` performance for a 1-dimensional test case, our uwGAN easily circumvents `Vegas` limitations from the assumed dimensional factorization.

As an LHC benchmark we use $\mu^+\mu^-$ production and a poor in-house event generator with a low unweighting efficiency over phase space. The uwGAN performs significantly better than the standard unweighting procedure, both, in kinematic tails and for noisy training data. Based on the success of our GAN architecture for top pair production [24] we expect our unweighting GAN to also work for higher final-state multiplicities. While it is not clear how much the speed gain from using an NN-unweighting in standard event generators will be, this application of generative networks could be easily implemented in the established LHC event generation chain. A challenge for any phase-space-related network are processes with a variable, llarge number of final-state particles, like $V$+jets production [15, 17]. While we were finalizing this study, similarly promising ideas were presented in Ref. [61], showing how generative networks benefit from training on weighted events.

### Acknowledgements

We want to thank Aishik Gosh and David Rousseau for inspiring this project. RW acknowledges support by the IMPRS-PTFS and by HeiKA. The research of AB and TP is supported by the Deutsche Forschungsgemeinschaft (DFG, German Research Foundation) under grant 396021762 — TRR 257 *Particle Physics Phenomenology after the Higgs Discovery*.

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
