# Peer review of "How to GAN Event Unweighting"

_SciPost Physics, doi:SciPost Phys. 10, 089 (2021)_

## Round 2 · Referee Report · Anonymous · 2021-2-19

Report

In this paper, the authors apply generative adversarial networks to the unweighting problem, a computational bottleneck in the production of high-precision calculations for the LHC experiments using Monte Carlo event generators.

The authors benchmark their setup against the mainstream VEGAS algorithm but also construct a deliberately challenging unweighting problem using Drell-Yan production as an example.

The study is very interesting and has clear relevance to the success of the LHC programme.

Would the authors have any insights as to what might be the origin of the small bias causing the uwGAN curve to be horizontally shifted with respect to the true distributions in Figure 1, at least where unweighted events are involved (left and right, but seemingly not in the centre)? A similar slope is also visible in the uwGAN/Truth ratio for the 2D Gaussian (Figure 3 bottom left), so it appears to be a systematic trend. It seems odd that the GAN would do so well with the complex toy models but fail to correct for such a comparatively straightforward shift?

Requested changes

The paper is overall rather well written, except for occasional machine-learning jargon that puts unnecessary obstacles in the path of readers not too well versed in the respective lingo. For example, it would improve the readability if abbreviations like "ReLU" or "ELU" were appropriately introduced in order to avoid the average reader having to consult Google. Similarly, spelling out "MMD" once would probably be beneficial.

  • validity: good
  • significance: high
  • originality: good
  • clarity: high
  • formatting: good
  • grammar: excellent

Author Ramon Winterhalder on 2021-03-25
(in reply to Report 1 on 2021-02-19)

-> First of all thank you to all three referees for their positive and
helpful comments!

The paper is overall rather well written, except for occasional
machine-learning jargon that puts unnecessary obstacles in the path of
readers not too well versed in the respective lingo. For example, it
would improve the readability if abbreviations like "ReLU" or "ELU"
were appropriately introduced in order to avoid the average reader
having to consult Google. Similarly, spelling out "MMD" once would
probably be beneficial.

-> We clarified the jargon, and noticed that there is no reason to
mention ELU at all.

---

## Round 2 · Referee Report · Anonymous · 2021-2-22

Report

The authors discuss the application of GANs to unweighting of event samples and perform a comparative study. I think this work is interesting and could inform the next generation of Monte Carlo event generation and their processing. I would encourage the authors to address one point before publication.

Drell-Yan is a very simple process and while their work shows that the GAN unweighting sufficiently reproduces the results of their event generator, the scalability of their results to more complex (and perhaps phenomenologically more relevant) final is not clear. Usually GANs (like other neural networks) can turn out to be fragile constructs when degrees of freedom change and complexity increases. Could the authors comment on the scalability of their framework? Do they expect any runtime improvements in more complex final states? Would this stand in tension with (perhaps excessive) process-dependent hyperparameter tuning?

  • validity: -
  • significance: -
  • originality: -
  • clarity: -
  • formatting: -
  • grammar: -

Author Ramon Winterhalder on 2021-03-25
(in reply to Report 2 on 2021-02-22)

-> First of all thank you to all three referees for their positive and
helpful comments!

Drell-Yan is a very simple process and while their work shows that the
GAN unweighting sufficiently reproduces the results of their event
generator, the scalability of their results to more complex (and
perhaps phenomenologically more relevant) final is not clear. Usually
GANs (like other neural networks) can turn out to be fragile
constructs when degrees of freedom change and complexity
increases. Could the authors comment on the scalability of their
framework? Do they expect any runtime improvements in more complex
final states? Would this stand in tension with (perhaps excessive)
process-dependent hyperparameter tuning?

-> Because we already know that we can GAN ttbar events we are
confident that this process will also work including the
unweighting. The true challenge will be processes with a variable
and large number of final state particles, like V+jets. We added
these comments also to the Outlook. Our hyper-parameter are not
very much tuned right now, and we assume that things will not get
much worse.

---

## Round 2 · Referee Report · Anonymous · 2021-2-23

Strengths

The paper explores a new approach to unweighting, which is an important part of MC event generation crucial to analysis of collider experiments.

Weaknesses

Some points need to be clarified - see report.

Report

This paper continues the exploration of GANs as a tool to improve Monte Carlo simulations of high-energy particle collisions. The general idea is to train a GAN on a weighted event sample produced by a standard first-principles event generator, and use GAN to produce an unweighted event sample which can then be directly compared with experimental data. This approach is an alternative to the standard unweighting procedure which may discard a significant fraction of events in the weighted sample, leading to inefficiencies.

Overall this is an interesting idea to try, and as far as I know it has not been tried before, making the paper worthy of publication. However I think there are a couple of points that require clarification.

1. Concerning the 500k Drell-Yan events shown in Figs. 4 and 5 and used in the discussion of sec. 3: are these events uniformly distributed in the 4D phase space defined by eq. (17)? If not, what is the distribution? The reason this is important is that any standard MC algorithm applied to DY would involve a phase space mapping which would greatly improve the unweighting efficiency of the standard unweighting procedure compared to uniform sampling. So, when the authors claim at the end of Sec. 3 that “a factor 100 between standard unweighting and the uwGAN method”, it is important to be more explicit about the precise meaning of “standard unweighting”.

2. While GAN algorithm can be used to generate large samples fast, fundamentally the GAN is an extrapolation of the training sample, so at some level its power must be limited by the size and quality of the training sample. If a physical observable is computed using the GAN-generated sample, in addition to purely statistical uncertainty, there will also be an uncertainty associated with training. (See for example 2002.06307). If the training sample is inefficient in terms of phase space coverage (e.g. contains too many events in areas with low cross section and not enough in areas with high cross section), it seems inevitable that this will ultimately limit the precision that can be obtained with GAN. Please comment.

Requested changes

The points listed in the report need to be addressed.

  • validity: top
  • significance: high
  • originality: high
  • clarity: high
  • formatting: perfect
  • grammar: perfect

Author Ramon Winterhalder on 2021-03-25
(in reply to Report 3 on 2021-02-23)

-> First of all thank you to all three referees for their positive and helpful comments!

  1. Concerning the 500k Drell-Yan events shown in Figs. 4 and 5 and used in the discussion of sec. 3: are these events uniformly distributed in the 4D phase space defined by eq. (17)? If not, what is the distribution? The reason this is important is that any standard MC algorithm applied to DY would involve a phase space mapping which would greatly improve the unweighting efficiency of the standard unweighting procedure compared to uniform sampling. So, when the authors claim at the end of Sec. 3 that “a factor 100 between standard unweighting and the uwGAN method”, it is important to be more explicit about the precise meaning of “standard unweighting”.

-> Yes, the events are distributed according to the 4D phase space defined in eq. (17). Indeed, in this simple case one could introduce a phase-space mapping which improves the unweighting efficiency significantly. However, our aim was to mimic the low unweighting efficiencies usually appearing in more involved mult-jet processes while having a simple and analytic expression for the amplitude and the phase space.

  1. While GAN algorithm can be used to generate large samples fast, fundamentally the GAN is an extrapolation of the training sample, so at some level its power must be limited by the size and quality of the training sample. If a physical observable is computed using the GAN-generated sample, in addition to purely statistical uncertainty, there will also be an uncertainty associated with training. (See for example 2002.06307). If the training sample is inefficient in terms of phase space coverage (e.g. contains too many events in areas with low cross section and not enough in areas with high cross section), it seems inevitable that this will ultimately limit the precision that can be obtained with GAN. Please comment.

-> We have shown in an earlier paper (2008.06545) that at least in principle a generative network can beat statistical limitations of a training sample, even though this has never really been analysed for phase-space GANs. The dijetGAN probably includes the most interesting hints in this direction. The unweighting GAN works differently, though, in the sense that it does not primarily benefit from the GAN interpolation, but it allows to unweight events without losing information. This means it should not be limited by interpolation systematics. We clarify this important point in the introduction, the discussion of Fig.5, and the outlook, thank you for bringing it up.

Anonymous on 2021-04-05
(in reply to Ramon Winterhalder on 2021-03-25)

Thanks for the clarifications. My comments have been addressed adequately and I recommend that this version of the paper be published.

---

## Round 3 · Referee Report · Anonymous (Referee 4) · 2021-3-29

Report

The authors have addressed my comments sufficiently. I am happy for publication to proceed.

---

## Round 3 · Referee Report · Anonymous (Referee 5) · 2021-4-7

Report

Dear authors,

thanks for clarifying the jargon in the new version.

Did you have any thoughts on my previous question as to where that small horizontal shift between the uwGAN curve and true distribution in the bottom left/right plots of Fig1 might come from (also visible as a slope in the bottom left plot of Fig3)?

Requested changes

No specific changes requested.

  • validity: -
  • significance: -
  • originality: -
  • clarity: -
  • formatting: -
  • grammar: -

Author:  Ramon Winterhalder  on 2021-04-12  [id 1353]

(in reply to Report 2 on 2021-04-07)

Did you have any thoughts on my previous question as to where that small horizontal shift between the uwGAN curve and true distribution in the bottom left/right plots of Fig1 might come from (also visible as a slope in the bottom left plot of Fig3)?

-> The mentioned small shift is not reproducible over various training runs. Minor deviations between the uwGAN
curve and the true distribution are always visible and are an effect of limited training size and limited statistics.

---

## Editorial Decision

published